The Tibetan medicine Zuotai influences clock gene expression in the liver of mice

Li Huan 1
Li Wen-Kai 1
Lu Yuan-Fu 1
Wei Li-Xin 2
Liu Jie 1 Jie@liuonline.com
1 Key Lab for Basic Pharmacology of Ministry of Education, Zunyi Medical College , Zunyi , China
2 Northwest Plateau Institute of Bology, Chinese Academia of Sciences , Xining, Qinghai , China
Rocha Joao
Electronic publication date: 2016 Jan 26
Publication date: 2016
Volume: 4
Electronic Location ID: e1632
Received 2015 Nov 4; Accepted 2016 Jan 6
Copyright: © 2016 Li et al.
Copyright year: 2016
Copyright holder: Li et al.
License: This is an open access article distributed under the terms of the Creative Commons Attribution License, which permits unrestricted use, distribution, reproduction and adaptation in any medium and for any purpose provided that it is properly attributed. For attribution, the original author(s), title, publication source (PeerJ) and either DOI or URL of the article must be cited.
License URL: https://creativecommons.org/licenses/by/4.0/

Keywords: Circadian clock, Zuotai, Npas2, Dbp, Liver, Bmal1, Nr1d1, Nfil3, Cry1, Per2

Funding: National Natural Science Foundation of China 81374063 Science and Technology Foundation of Guizhou Province 2010-C-527; 2011-5086 Key Laboratory Special Development Program of Qinghai Province 2014-Z-Y02 This study was supported by the National Natural Science Foundation of China (81374063), the Science and Technology Foundation of Guizhou Province (2010-C-527; 2011-5086) and Key Laboratory Special Development Program of Qinghai Province (2014-Z-Y02). The funders had no role in study design, data collection and analysis, decision to publish, or preparation of the manuscript.

==============================
Background. The circadian clock is involved in drug metabolism, efficacy and toxicity. Drugs could in turn affect the biological clock as a mechanism of their actions. Zuotai is an essential component of many popular Tibetan medicines for sedation, tranquil and “detoxification,” and is mainly composed of metacinnabar (β-HgS). The pharmacological and/or toxicological basis of its action is unknown. This study aimed to examine the effect of Zuotai on biological clock gene expression in the liver of mice. Materials and methods. Mice were orally given Zuotai (10 mg/kg, 1.5-fold of clinical dose) daily for 7 days, and livers were collected every 4 h during the 24 h period. Total RNA was extracted and subjected to real-time RT-PCR analysis of circadian clock gene expression. Results. Zuotai decreased the oscillation amplitude of the clock core gene Clock, neuronal PAS domain protein 2 (Npas2), Brain and muscle Arnt-like protein-1 (Bmal1) at 10:00. For the clock feedback negative control genes, Zuotai had no effect on the oscillation of the clock gene Cryptochrome (Cry1) and Period genes (Per1–3). For the clock-driven target genes, Zuotai increased the oscillation amplitude of the PAR-bZip family member D-box-binding protein (Dbp), decreased nuclear factor interleukin 3 (Nfil3) at 10:00, but had no effect on thyrotroph embryonic factor (Tef); Zuotai increased the expression of nuclear receptor Rev-Erbα (Nr1d1) at 18:00, but had little influence on the nuclear receptor Rev-Erbβ (Nr1d2) and RORα. Conclusion. The Tibetan medicine Zuotai could influence the expression of clock genes, which could contribute to pharmacological and/or toxicological effects of Zuotai.

Introduction

Traditional Tibetan medicine is one of the four traditional medicines in the world and has a unique theoretical system and an ability to make diagnosis and treatment of diseases. Many Tibetan medicines contain Zuotai (Kan, 2013), a mixture of metal ash. Modern analytical methods show that Zuotai contains mainly β-HgS, and other trace elements (Li et al., 2015). In experimental animals, Zuotai at 4.5-fold of clinical dose (30 mg/kg) did not show overt toxicity towards the kidney and liver as compared to HgCl2 (at equivalent Hg dose) or MeHg (at 1/10 Hg dose) (Lu et al., 2015). In a recent clinical trial, Zuotai-containing Danzuo did not show obvious adverse effects in patients under the clinical doses and duration of administration (Li et al., 2014). Pharmacology studies have shown that Zuotai has the effects of sedation (tranquil), anti-inflammation, modulation of the immune system, and prolongs the life of fruit flies (Huang et al., 2013; Kan, 2013). Repeated administration of Zuotai (4–12 mg/kg, po for 12 days) in rats could affect the activity, protein and mRNA expression of CYP1A2 and N-acetyltransferase 2 (Li et al., 2014). The recent researches on mercury sulfide (α-HgS, β-HgS)-based traditional medicines, either from Ayurvedic medicine, Tibetan medicine, or Chinese medicine, have been reviewed (Kamath et al., 2012; Chen et al., 2012). However, the actions and mechanisms of pharmacological and toxicological effects of Zuotai remain unclear; more studies are needed to provide the scientific basis for this traditional medicine.

Chronopharmacology emerges as novel targets of therapeutics and drug safety (Dallmann, Brown & Gachon, 2014). The circadian timing system not only rhythmically controls behavior, physiology, cellular proliferation over the 24-h period (Mohawk, Green & Takahashi, 2012; Richards & Gumz, 2013), but also implicates in drug metabolism, efficacy, toxicity and detoxification (Bailey, Udoh & Young, 2014; DeBruyne, Weaver & Dallmann, 2014; Zmrzljak & Rozman, 2012). In mammals, the mechanism of the circadian clock is regulated by delicate systems. At the core of this clock network are the transcriptional activators, Clock and its paralog neuronal PAS domain protein 2 (Npas2), Brain and muscle Arnt-like protein-1 (Bmal1), positively regulate the expression of the Period genes (Per1, Per2 and Per3) and Cryptochrome genes (Cry1, Cry2) at the beginning of the cycle. Per and Cry gene products accumulate, dimerize, and form a complex to interact with Clock-Bmal1, repressing their own transcription (Mohawk, Green & Takahashi, 2012). Clock-Bmal1 activate the nuclear orphan receptor protein Rev-Erbα (Nr1d1) gene, and the PAR-bZip family members such as D-box-binding protein (Dbp), thyrotroph embryonic factor (Tef), nuclear factor interleukin 3 (Nfil3), all of which are transcriptional targets of CLOCK-BMAL1 (Mohawk, Green & Takahashi, 2012) affecting drug metabolism and detoxification (Dallmann, Brown & Gachon, 2014).

Accumulating evidence demonstrated that circadian clock could be altered by drugs and toxicants. For example, hepatic fibrosis induced by carbon tetrachloride in mice leads to alterations in the circadian rhythms of hepatic clock genes (Chen, Kakan & Zhang, 2010). Acetaminophen hepatotoxicity is also influenced by clock gene Per2 (Kakan, Chen & Zhang, 2011). Circadian clock genes are altered in livers of chronic ethanol-fed mice (Filiano et al., 2013). Dioxin induction of Cyp1a1 is influenced by period gene expression (Qu et al., 2009). Thus, drugs and toxicants could affect the circadian rhythm as a mechanism of their toxicity.

Therapeutic agents could also affect circadian clock to exert their beneficial effects. For example, resveratrol reverses high-fat diet induced circadian disruption (Miranda et al., 2013). Dietary oleanolic supplementation affects clock gene expression to produce beneficial effects (Gabas-Rivera et al., 2013). The antidiabetic drug metformin modulate the positive loop of the circadian clock (Barnea et al., 2012). Therefore, the goal of this study is to investigate the effect Zuotai on peripheral circadian clock in livers of mice in an attempt to gain new insights into the therapeutic basis and toxicity of this traditional medicine.

Materials and Methods

Animals and chemicals

Male outbreed Kunming mice (6 weeks of age) were purchased from the Experimental Animal Center of Third Military Medical College (Chongqing, China) and acclimatized for one week before experiments. Mice had free access to rodent chow and drinking water in the SPF-grade animal facilities with 21 ± 2 °C and the light is from 8:00 to 20:00. All animal procedures follow the NIH guide of Humane Use and Care Animals, and were approved by Institutional Animal Use and Care Committee of Zunyi Medical College (2014–07). Zoutai was obtained from the Institute of Northwest Plateau Biology, Chinese Academy of Sciences (Li et al., 2014; Li et al., 2015).

Animal treatment

Mice were divided into 12 groups (n = 5) randomly. Six groups of mice were orally administrated with Zuotai at the dose of 10 mg/kg (1.5-fold of clinical dose), for 7 days in the morning; control mice received the same volume (10 ml/kg) of saline. One day after the last dose, mice were anesthetized with 7% chloralhydrate and liver tissues were harvested at 10:00, 14:00, 18:00, 22:00, 02:00, and 06:00, respectively. Livers were kept in −80 °C prior to analysis.

RNA isolation

Approximately 50–100 mg liver tissues were homogenized in 1ml Trizol (TakaRa Biotechnology, Dalian, China). The quality and quantity of RNA were determined by NanoDrop 2000 Spectrophotometer (Thermo Scientific, Waltham, MA, USA). Total RNA was reversed transcribed into cDNA with TakaRa RT kits (Dalian, China).

Real-time RT-PCR analysis

The primers were designed with Primer3 software and listed in Table 1.

Table 1 Primer sequences for real-time RT-PCR.

Gene	Access	Forward	Reverse	
β-actin	NM_031144	TTGCCCTAGACTTCGAGCAA	CAGGAAGGAAGGCTGGAAGA	
Bmal1	NM_024362	TGAACCAGACAATGAGGGCT	TATGCCAAAATAGCCGTCGC	
Clock	NM_021856	CTCCCCACAAGACTGCAGTA	CCTGTGTGGCCTTTACCCTA	
Cry1	NM_198750	TACAGCAGCCACAAACAACC	TCCTGACGAAGCTGTGTCAT	
Dbp	NM_012543	CCAGTGCTCCTGGCATGACTAA	GCCTTCACAAGCATGAACTCCATA	
Per1	NM_001034125	TGAGCTCATGAACCTGGGAG	TCTTTGGGCTTGCTGTTTCC	
Per2	NM_031678	GTCCCCGGCTAGAAGTCTAC	TAAACCTCCCCACAGCTCTG	
Per3	MA164628	CTCAAGACGTGAGGGCGTTCTA	GGTTTCGCTGGTGCACATTC	
Nfil3	MA059139	GGTTACAGCCGCCCTTTCTTT	AAGGACTTCAGCCTCTCATCCATC	
Nr1d1	NM_001113422	AGCTGGTGAAGACATGACGA	GGTGGGAAGTATGTGGGACA	
Nr1d2	MA030409	CCAGTGCTCCTGGCATGACTAA	GCCTTCACAAGCATGAACTCCATA	
Npas2	MA151656	TGCTCCGAGAATCGAATGTGATA	ATGGCAGGCTGCTCAGTGAA	
ROR-α	NM_001289917	GAACCTTGCCTTTGGACCTG	TGGAGCTGGACTAGAGGT	
Tef	MA032354	CTTCAACCCTCGGAAGCACA	CCGGATGGTGATCTGGTTCTC	

The IQTM SYBR Green Supermix (Bio-Rad Laboratories, Hercules, CA, USA) was used for real time RT-PCR analysis. The 15 μl reaction mix contained 3 μl of cDNA (10 ng/μl), 7.5 μl of SYBR Green (2×), 0.5 μl of primer mix (10 μM each), and 4 μl of ddH2O. After 5 min denature at 95 °C, 40 cycles will be performed: annealing and extension at 60 °C for 45 seconds and denature at 95 °C for 10 seconds. Dissociation curve was performed after finishing 40 cycles to verify the quality of primers and reaction. The expression of genes was calculated by the 2−ΔΔCt method (Schmittgen & Livak, 2008). The housekeeping gene β-actin was used for normalization.

Statistical analysis

All data are given as mean ± standard error of the mean (SEM). The peak/tough ratios during the 24 hr period were calculated for oscillation amplitude comparison. Student’s t test was performed to compare the gene expression levels between control and Zuotai group at each time point. P < 0.05 was set as the criteria of significance.

Results

Clock master control genes

The clock master control genes include Clock, Npas2 and Bmal1 (Mohawk, Green & Takahashi, 2012; Richards & Gumz, 2013). Effects of Zuotai on the expression of clock master control genes Clock, Npas2 and Bmal1 are shown in Fig. 1. The clock core regulation genes Clock, Npas2 and Bmal1 displayed typical circadian oscillation patterns in the control group. Clock had a downward trend from 10:00 to 18:00, a rising trend from 18:00 to 10:00. The peak/tough ratio for Clock was 6.3 in control group but is was 2.9 in Zuotai group; at 10:00, the Clock mRNA levels were lower in Zuotai group. Nasp2, as a paralog of core clock gene Clock, increased from 2:00 to 10:00 and decreased gradually from 10:00 to 2:00. The peak/tough ratio for Npas2 was 253 in control group but is was 102 in Zuotai group; Bmal1 rapidly declined from 6:00 to 18:00 and increased from 18:00 to 6:00. The peak/tough ratio for Bmal1 was 50 in control group but is was 10 in Zuotai group; at 10:00, the Bmal1 mRNA levels were lower in Zuotai group. Zuotai appeared to decrease the expression of these clock master genes.

Figure 1 Effects of Zuotai on the expression of clock master control gene Clock, Npas2 and Bmal1.

Mice were given the dose of Zuotai (10 mg/kg, po) for 7 days, and the livers were collected at 6:00, 10:00; 14:00, 18:00, 22:00 and 2:00. Total RNA was extracted and subjected to real-time RT-PCR analysis. Data are the mean and SEM of 5 mice. The values in parentheses represent the peak/tough ratio during the 24 hr period. *Significantly different from controls p < 0.05.

Clock feedback control genes

The clock feedback regulation genes mainly consist of Per1, Per2, per3 and Cry1, Cry2 (Mohawk, Green & Takahashi, 2012; Richards & Gumz, 2013). As shown in Fig. 2, there were four feedback regulation genes Per1, Per2, per3 and Cry1, which are activated directly by the dipolymer BMAL1-CLOCK. All the four feedback genes displayed typical circadian oscillation patterns in the control group. Per1 and Per2 (Fig. 2) both were upward from 14:00 to 22:00 and downward from 22:00 to 14:00. The peak/tough ratio was 8.0 for Per1 in control group and 8.2 in Zuotai group; The peak/tough ratio was 5.4 for Per2 in control group and 17 in Zuotai group; Per3 raised from 10:00 to 18:00 and declined from 18:00 to 10:00, the peak/tough ratio was 39 in control group and 35 in Zuotai group; Cry1 decreased straightly from 6:00 to 18:00, and from 18:00 to 6:00, it increased gradually, with the peak/tough ratio of 4.7 in control group and 4.7 in Zuotai group. Zuotai had little effects on the circadian rhythm of these feedback control genes.

Figure 2 Effects of Zuotai on the expression of the clock feedback control gene Per1, Per2 (left) and Per3 145 and Cry1 (right). Mice were given the dose of Zuotai (10 mg/kg, po) for 7 days, and the livers were collected at 6:00, 10:00; 14:00, 18:00, 22:00 and 2:00.

Total RNA was extracted and subjected to real-time RT-PCR analysis. Data are the mean and SEM of 5 mice. The values in parentheses represent the peak/tough ratio during the 24 hr period.

Clock targeted and/or driven genes

The clock targeted/driven genes include Nfil3, Tef, RORα (Bailey, Udoh & Young, 2014; DeBruyne, Weaver & Dallmann, 2014; Zmrzljak & Rozman, 2012). Dbp, Nr1d1, and Nr1d2 are also lock-driven genes, and Nr1d1 can also negatively regulate the master core clock genes Bmal1 and Clock (Mohawk, Green & Takahashi, 2012; Dallmann, Brown & Gachon, 2014). In Fig. 3, the clock targeted genes Nfil3 and Tef show circadian rhythm with the variation of time. From 18:00 to 10:00, Nfil3 tended to be peaked at 10:00. For Tef, it peaked at 18:00. The peak/tough ratio for Nfil3 was 11 in control group bur it was 6.2 in Zuotai group, and at 10:00, the Nfil3 mRNA levels were lower in Zuotai group; the peak/tough ratio for Tef was 13 in control group bur it was 6.7 in Zuotai group. Although Zuotai decreased the expression of Nril3 at 10:00, it had little effects on the circadian rhythm of Tef.

Clock-driven gene Dbp, Nr1d1, and Nr1d2 displayed typical circadian oscillation patterns peaked around 18:00. The peak/tough ratio for Dbp was 12 in control group but it was 58 in Zuotai group; the peak/tough ratio Nr1d1 was 54 in control group but it was 89 in Zuotai group, and at 18:00, the Nr1d1 mRNA levels were significantly higher in Zuotai group. The peak/tough ratio for Nr1d2 was 10 in control group and it was 8.3 in Zuotai group. For RORa, it increased from 22:00 to 6:00 and fell from 6:00 to 22:00, the peak/tough ratio for RORa was 2.6 in control group and it was 2.2 in Zuotai group. Zuotai had little effects on the circadian rhythm of Nr1d2 and RORa.

Figure 3 Effects of Zuotai on the expression of Clock targeted/driven genes Nfil3, Tef, Dbp, Nr1d1, Nr1d2, and RORα.

Mice were given the dose of Zuotai (10 mg/kg, po) for 7 days, and the livers were collected at 6:00, 10:00; 14:00, 18:00, 22:00 and 2:00. Total RNA was extracted and subjected to real-time RT-PCR analysis. Data are the mean and SEM of 5 mice. The values in parentheses represent the peak/tough ratio during the 24 hr period. *Significantly different from controls p < 0.05.

Discussion

The present study demonstrated that the liver of Kunming mice showed typical circadian rhythm as C57mice. Generally speaking, Zuotai did not markedly disrupt the intrinsic circadian rhythm of Per1, Per3, Cry1, Nr1d2 and RORα, but it attenuated oscillation of Bmal1, Clock, Npas2, and increased oscillation of Dbp and Nr1d1 in the liver of mice. This is the first study on the potential influence of Tibetan medicine Zuotai on hepatic clock gene expression.

Circadian clock system consists of central clock and peripheral clock (Mohawk, Green & Takahashi, 2012). The central clock is located in the suprachiasmatic nuclei (SCN) of the hypothalamus, and is regulated by light, feeding cues and temperature cycle (Fuhr et al., 2015). The peripheral clocks reside in various tissues throughout the body. The peripheral clocks play an integral and unique role in respective tissues, driving the circadian expression of specific genes involved in a variety of physiological functions (Richards & Gumz, 2013). It is well established that peripheral clock was involved in carbohydrate metabolism, lipid metabolism, protein and amino acid metabolism (Bailey, Udoh & Young, 2014), especially in process of drug metabolism, including absorption from the gastro-intestinal tract, biotransformation in the liver, and hepatobiliary excretion (Dallmann, Brown & Gachon, 2014). Thus, peripheral clock is important as central clock in pharmacology and toxicology.

Circadian clock can be disrupted by drugs and toxicants. For example, Ecstasy (MDMA) reduced expression of Bmal1, Clock, and Npas2 in the heart of C57 mice following repeated administrations (Koczor et al., 2015). Ethanol-induced hepatotoxicity is influenced by clock gene Per1, and deletion of Per1 protects mice from ethanol-induced liver injury by decreasing hepatic lipid accumulation (Wang et al., 2013). Chronic ethanol consumption disrupts several metabolic pathways including β-oxidation and lipid biosynthesis, and disrupts the diurnal oscillations of core clock genes (Bmal1, Clock, Cry1, Cry2, Per1, and Per2), and disrupts the expression of clock-controlled genes Dbp, Hlf, Nocturnin, Npas2, Rev-erba, and Tef (Filiano et al., 2013). Clock genes also affect the cytotoxicity of diethylnitrosamine (DEN), possibly by affecting the bioactivation of DEN and by inducing apoptosis (Matsunaga et al., 2011), and DEN-induced hepatocarcinogenesis is associated with disruption of clock genes Bmal1, Dbp and Rev-Erba (Jin et al., 2013). Clock gene Per2 functions in diurnal variation of acetaminophen induced hepatotoxicity via modulating Cyp1a2 expression in mice (Kakan, Chen & Zhang, 2011). Circadian clock also controls acetaminophen bioactivation through NADPH-cytochrome P450 oxidoreductase (Johnson et al., 2014). Circadian clock disruption is also involved in CCl4-induced chronic liver fibrosis (Chen et al., 2009; Chen, Kakan & Zhang, 2010). Thus, disruption of peripheral clock is a novel target of toxic effects of chemicals. Whether the alteration of circadian clock could be related to toxicity potential of Zuotai requires further investigation.

Many drugs could alter circadian clock to exert their therapeutic effects. For example, Resveratrol reverses the change induced by high-fat feeding in the expression of Rev-Erba in adipose tissue, which means that clock machinery is a target for this polyphenol (Miranda et al., 2013). Oleanolic acid is a triterpenoid widely distributed throughout the plant kingdom and has many beneficial effects (Liu et al., 2008). Dietary oleanolic acid supplementation (0.01%) for 11 weeks increased Bmal1 and Clock gene expression (Gabas-Rivera et al., 2013). The antidiabetic drug metformin resulted in a decrease in Bmal1 expression, but an increase in Clock expression in the liver of C57BL/6 male mice. Metformin also led to the activation of liver casein kinase Iα (CKIα) and muscle CKIɛ, known modulators of the positive loop of the circadian clock (Barnea et al., 2012). Dietary lipoic acid supplementation could up-regulate circadian genes in the positive arm (Bmal1 and Npas2, a functional homologue of the Clock gene) and down-regulate genes in the negative arm (Per2, Per3, Nr1d2) of the circadian core oscillators (Finlay et al., 2012). Bavachalcone, a natural medicine ingredient, has a pharmacological function in regulating RORα (Dang et al., 2015). Thus, alteration of circadian clock could be a pharmacological basis of therapeutics.

We recently examined the circadian and sex variations of liver detoxification components such as Nrf2 (Xu et al., 2012), metallothionein (Zhang et al., 2012), as well as cytochrome P450 enzyme genes (Lu et al., 2013). Our results demonstrate that the peripheral clock is equally important to the central clock in pharmacology. Indeed, drugs and toxicants (such as alcohol) could affect peripheral clock without affecting central clock at SCN to produce biological effects (Filiano et al., 2013; Musiek & Fitzgerald, 2013).

The present study extended our efforts in the study of the Tibetan medicine Zuotai, from chemical analysis of Zuotai components (Li et al., 2015), animal toxicity study of Zuotai and clinical safety evaluation of Zuotai-containing Tibetan medicine Danzuo (Li et al., 2014), and the dissolution, absorption and bioaccumulation in gastrointestinal tract (Zheng et al., 2015).

In comparison with HgCl2, Zuotai is much less dissolved, absorbed, accumulated in the liver, and produces much less hepatotoxicity and nephrotoxicity as compared to HgCl2 or MeHg (Zheng et al., 2015; Lu et al., 2015). Whether the changes in the expression of circadian clock genes is related to toxicity or the therapeutic effects of Zuotai need further investigation.

In summary, the present studies demonstrate that the Tibetan medicine Zuotai at the clinical, non-toxic dose could decrease the oscillation of the core clock Bmal1, Clock and Npas2, increase the oscillation of the clock driven genes Dbp and Nr1d1, while it has no effects on the circadian feed control gene Per1, Per2, Per3 and Cry1, as well as Tef, Nr1d2 and RORα. These results could provide new insights and add our understanding of pharmacological and/or toxicological actions of the Tibetan medicine Zuotai.

Supplemental Information

Supplemental Information 1 X cell raw data.

Nfil3 raw data.

Click here for additional data file.

Supplemental Information 2 X cell raw data.

RORa raw data.

Click here for additional data file.

Supplemental Information 3 X cell raw data.

Nr1d2 raw data.

Click here for additional data file.

Supplemental Information 4 X cell raw data.

Per3 raw data.

Click here for additional data file.

Supplemental Information 5 X cell raw data.

Per1 raw data.

Click here for additional data file.

Supplemental Information 6 X-cell raw data.

Bmal 1 raw data.

Click here for additional data file.

Supplemental Information 7 X cell raw data.

Cry1 raw data.

Click here for additional data file.

Supplemental Information 8 X cell raw data.

Clock raw data.

Click here for additional data file.

Supplemental Information 9 X cell raw data.

Nr1d1 raw data.

Click here for additional data file.

Supplemental Information 10 X cell raw data.

Dbp raw data.

Click here for additional data file.

Supplemental Information 11 X cell raw data.

Per2 raw data.

Click here for additional data file.

Supplemental Information 12 X cell raw data.

Npas2 raw data.

Click here for additional data file.

Supplemental Information 13 X cell raw data.

Tef raw data.

Click here for additional data file.

Additional Information and Declarations

Competing Interests

Author Contributions

Animal Ethics

Data Deposition

Jerry (Jie) Liu is an Academic Editor for PeerJ.

Huan Li conceived and designed the experiments, performed the experiments, analyzed the data, wrote the paper, prepared figures and/or tables.

Wen-Kai Li performed the experiments, analyzed the data.

Yuan-Fu Lu conceived and designed the experiments, analyzed the data, contributed reagents/materials/analysis tools, reviewed drafts of the paper.

Li-Xin Wei analyzed the data, contributed reagents/materials/analysis tools, reviewed drafts of the paper, provided test materials.

Jie Liu conceived and designed the experiments, performed the experiments, analyzed the data, wrote the paper, prepared figures and/or tables.

The following information was supplied relating to ethical approvals (i.e., approving body and any reference numbers):

All animal procedures follow the NIH guide of Humane Use and Care Animals, and were approved by Institutional Animal Use and Care Committee of Zunyi Medical College (number: 2014–07).

The following information was supplied regarding data availability:

Data can be found in the Supplemental Information.

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
