# Peer review of "The Tibetan medicine Zuotai influences clock gene expression in the liver of mice"

_PeerJ, doi:10.7717/peerj.1632_

## Round 0.1 · original submission · Major Revisions

· Academic Editor

Major Revisions

Dear Dr Liu,

Find attached the comments about your manuscript. The reviewers have raised several important points to be considered by you. In particular, after reading the manuscript, I also recommend you to include both in abstract, introduction and discussion aspects related to the potential toxicity of the preparation. The fact that the preparation is used in folk medicine does not guarantee its safety and the presence of Hg (and other toxic metals) in the preparation is worrying.

Reviewer 1 ·

Basic reporting

Minor Points
1. The author should present some reference in the description of real-time RT-PCR analysis, as well describe with more detail the technique.
2. In the results section some information about the gene mentioned should be added.
3. Language mistakes should be revised.

Experimental design

Major points
1. The project being presented here is quite confusing and more convincing way to present the issue is required.

Validity of the findings

Major points
2. The authors could describe the statistical analysis used to compare the results from control animals and animals treated with Zuotai. For instance, in the Figure 1, the author have to show the p value and statistical test that was used to compare the gene expression of Clock in the different periods in the different groups.
3. What changes in the metabolism and other parameters could be expected after the decreased in the circadian oscillation of Clock, Npas2, Bma11 liver expression induced by Zuotai treatment?
4. The authors should discuss more about the possible effects caused by the changes induced by Zuotai treatment in the expression of these different genes.
5. Some hormonal analysis in the mice’s plasma should be done, such as melatonin and corticosterone, in attempt to delve the mechanisms involved in the effects in gene expression induced by Zuotai treatment.
6.Its not clear that these alterations caused by the treatment are related to toxicological or therapeutics effects. The authors should clarify this point in the discussion section.

Additional comments

Synopsis
This is an interesting paper regarding the impact of Tibetan medicine Zuotai on liver biological clock in mice. The main finding is that Zuotai treatment modulates the expression of clock genes.

My comments
Some aspects need to be clarified.

Annotated reviews are not available for download in order to protect the identity of reviewers who chose to remain anonymous.

·

Basic reporting

There are text boxes on the text, it must be removed.
Some phrases are written in a confusing way and with many repeated words.
some phrases seem to have no connection with the next sentence of paragraph.

Experimental design

No Comments

Validity of the findings

It must perform an appropriate statistical test in the samples, since the average of the standard errors was very high, and thus, we can not trust these results.

Additional comments

No Comments

---

## Round 0.2 · Major Revisions

· Academic Editor

Major Revisions

Although part of the criticisms of the reviewers were answered by the authors, they have not properly addressed two questions raised by the reviewers. I, particularly, concur with the reviews about the toxicity. In addition, the statistic done by the authors was not the one adequate to the problem. One is the question of toxicity and the other is the statistical analysis used. Indeed, the major problem with the revised version is that the authors are not highlighting the potential toxicity of the use of the preparation used to intoxicated the rats. The fact that pattern of gene expression changed is a clear indication of potential toxicity (or even of toxicity). Thus, without clearly indicating the potential toxicological effects of the preparation and, in particular of cinnabar, the paper cannot be accepted. This can give the erroneous impression that the preparation is safe and, from a conservative point of view of toxicology (that any departure from the normal is potentially toxic) the results indicate that it is not safe. In fact, any deviation of a "normal condition" (here the control group gene expression)" can be an important indication of toxicity. Authors have to change the affirmation in the conclusion that the changes in the pattern of gene expression may be of pharmacological-therapeutic significance. Here, without the support of solid experimental evidence that safe medicines can exert their therapeutic effects by changing the expression of clock genes, the supposition made by the authors are very speculative.

Thus, in abstract, introduction and discussion, authors have to clearly indicate and cite literature (introduction and discussion) indicating the toxicity of cinnabar or preparations having it. For instance (and this is only a small list selected from the literature),

Kang-Yum, E., & Oransky, S. H. (1992). Chinese patent medicine as a potential source of mercury poisoning. Veterinary and human toxicology, 34(3), 235-238.

Chuu, J. J., Liu, S. H., & Lin-Shiau, S. Y. (2001). Effects of methyl mercury, mercuric sulfide and cinnabar on active avoidance responses, Na+/K+-ATPase activities and tissue mercury contents in rats. Proceedings of the National Science Council, Republic of China. Part B, Life sciences, 25(2), 128-136.

Seok, J., Park, K. Y., Li, K., Kim, B. J., Shim, J. H., Seo, S. J., ... & Hong, C. K. (2015). Squamous Cell Carcinoma and Multiple Bowen's Disease in a Patient with a History of Consumption of Traditional Chinese Herbal Balls. Case reports in dermatology, 7(2), 151-155.

Espinoza, E. O., Mann, M. J., & Bleasdell, B. (1995). Arsenic and mercury in traditional Chinese herbal balls. New England Journal of Medicine, 333(12), 803-804.

Chun-Fa Huang, Chuan-Jen Hsu, Shing-Hwa Liu, and Shoei-Yn Lin-Shiau, “Exposure to Low Dose of Cinnabar (a Naturally Occurring Mercuric Sulfide (HgS)) Caused Neurotoxicological Effects in Offspring Mice,” Journal of Biomedicine and Biotechnology, vol. 2012, Article ID 254582, 12 pages, 2012. doi:10.1155/2012/254582

Wei L, Liao P, Wu H, Li X, Pei F, Li W, Wu Y. Toxicological effects of cinnabar in rats by NMR-based metabolic profiling of urine and serum. Toxicology and applied pharmacology. 2008 Mar 15;227(3):417-29.

Yu WH, Zhang N, Qi JF, Sun C, Wang YH, Lin M. Arsenic and Mercury Containing Traditional Chinese Medicine (Realgar and Cinnabar) Strongly Inhibit Organic Anion Transporters, Oat1 and Oat3, In Vivo in Mice. BioMed Research International. 2015 Dec 16;2015.

Emslie SD, Brasso R, Patterson WP, Valera AC, McKenzie A, Silva AM, Gleason JD, Blum JD. Chronic mercury exposure in Late Neolithic/Chalcolithic populations in Portugal from the cultural use of cinnabar. Scientific reports. 2015;5.

The list is a random sample made and does not reflect all the studies presented. The statement that at therapeutic doses "the preparation containing cinnabar" is non-toxic can be misleading, because living cells do not need Hg and even small doses of Hg can be toxic in long-term exposure. Thus, without warning the readers about the potential toxicity of mercury, the study is biased, particularly, in view that even the "supposed safe cinnabar's containing preparations" can be misused.

A second critical point is that authors have not performed an adequate statistical test. As stated by the authors, the profile or the pattern of expression, can be more indicative than isolated comparisons. However, to state that one pattern of expression is different from the other, the authors must perform a TWO-WAY ANOVA (two treatments x times of sampling). A significant treatment x time interaction will indicate whether or not the treatment changed the pattern of gene expression.

---

## Round 0.3 · accepted · Accept

· Academic Editor

Accept

Dear Dr Liu, thank you for the revision. I am glad to accept the revised version of your manuscript.